# Classification of Microbial Activity and Inhibition Zones Using Neural Network Analysis of Laser Speckle Images

**DOI:** 10.3390/s25113462

**Published:** 2025-05-30

**Authors:** Ilya Balmages, Dmitrijs Bļizņuks, Inese Polaka, Alexey Lihachev, Ilze Lihacova

**Affiliations:** 1Institute of Applied Computer Systems, Riga Technical University, LV-1048 Riga, Latvia; dmitrijs.bliznuks@rtu.lv; 2Institute of Information Technology, Riga Technical University, LV-1048 Riga, Latvia; inese.polaka@rtu.lv; 3Institute of Atomic Physics and Spectroscopy, Faculty of Science and Technology, University of Latvia, LV-1040 Riga, Latvia; aleksejs.lihacovs@lu.lv (A.L.); ilze.lihacova@lu.lv (I.L.)

**Keywords:** laser speckle imaging, correlation analysis, image processing, signal processing, microorganism spatiotemporal activity estimation, classification of microorganism’s activity, artificial neural networks

## Abstract

This study addresses the challenge of rapidly and accurately distinguishing zones of microbial activity from antibiotic inhibition zones in Petri dishes. We propose a laser speckle imaging technique enhanced with subpixel correlation analysis to monitor dynamic changes in the inhibition zone surrounding an antibiotic disc. This method provides faster results compared to the standard disk diffusion assay recommended by EUCAST. To enable automated analysis, we used machine learning algorithms for classifying areas of bacterial or fungal activity versus inhibited growth. Classification is performed over short time windows (e.g., 1 h), supporting near-real-time assessment. To further improve accuracy, we introduce a correction method based on the known spatial dynamics of inhibition zone formation. The novelty of the study lies in combining a speckle imaging subpixel correlation algorithm with ML classification and with pre- and post-processing. This approach enables early automated assessment of antimicrobial effects with potential applications in rapid drug susceptibility testing and microbiological research.

## 1. Introduction

Laser speckle techniques enable the observation of fine details that are not visible under white light illumination. This method offers a fast, cost-effective, and non-contact method for assessing microbial activity. If the scatterers on the surface are stationary and static, the scattered light forms stable laser speckle patterns. However, if the scatterers move or change over time, time-varying speckles are produced [1]. The literature data indicate advances and potential applications of laser speckle techniques to evaluate dynamic processes in microbiological media. Applications include evaluating bacterial chemotactic response in agar plates [2] and distinguishing motile bacteria from fungi [3]. Technology using speckle decorrelation time maps has been demonstrated for the detection of *E. coli* and *B. cereus* on meat (chicken breast) [4]. Speckle analysis has also been applied to biomass growth kinetic measurements in liquid culture [5], characterization of CFU morphology [6], and determination of antibiotic susceptibility [7,8].

To validate laser speckle as a reliable method for monitoring microbial concentration, the literature suggests analyzing: the speckle grain size [9], the spatial contrast [10], temporal contrast, and a combination of both, spatio-temporal contrast [11]. As scatterer concentration in the sample increases, the dynamics of Brownian motion also increase. The speckle grain size is a parameter affected by the number or the size of the scatterers in the medium [12]. The demonstrated results in [5] show that both speckle grain size and spatial contrast decrease with microorganism growth. Another useful approach to using the laser speckle technique to determine microorganism activity is to measure the time or rate of decorrelation [4,13]. The decreases in the temporal correlation value are proportional to the microorganism’s concentration and activity. For analyzing speckle patterns containing both dynamic and static regions, study [14] demonstrated that laser speckle correlation can be effectively used to estimate the relative concentrations of static and dynamic scatterers within a sample. In our previous study [15], these methods were compared with the proposed subpixel correlation algorithm, which has proven to be a worthy tool for detecting microbial growth activity and detecting hidden effects in microorganism behavior. This method is based on the principle that deformation, vibration, or displacement of a surface induces measurable shifts in the speckle pattern [16], which can be quantified via the cross-correlation peak position shift between consecutive frames. By analyzing the resulting time signals, dynamic surface events, such as microbial growth, can be detected. The results and the advantages of the algorithm were demonstrated.

The laser speckle imaging technique has been proven to monitor moving particles (such as bacterial or fungal activity) in optically inhomogeneous media by analyzing time-varying laser speckle patterns using the subpixel correlation method [17].

It has also been demonstrated that by relying on differences in signal behavior, it is possible to distinguish (or classify) growing and non-growing bacterial colonies [18]. Accordingly, using the proposed properties, it is possible to distinguish active and inactive (inhibition) zones. Rapid classification into active and inhibited zones is important, as it allows the detection of the influence of chemical agents, such as drugs and antibiotics, on microorganism growth dynamics, thus speeding up and facilitating epidemiological analysis. This can provide targeted pharmacological intervention at the early stages of the disease, increasing patient survival chances.

To automate the classification between inhibition and active bacterial growth zones, various algorithmic tools can be utilized. These tools may include linear classifiers, logistic regression, quadratic classifiers, and Bayes classifiers, among others. Linear models are versatile and effective across a broad range of applications [19]. Speckle images contain complex spatiotemporal dynamics of bacterial activity and antibiotic response. Traditional statistical methods, such as speckle contrast, decorrelation, and subpixel correlation analysis, capture overall dynamic activity but poorly distinguish between close or overlapping classes. Machine learning, especially neural networks, effectively processes such data, identifying nonlinear relationships and providing high classification or prediction accuracy. Study [20] compares various machine learning classifiers, including support vector machine (SVM) [21], logistic regression (LR) [22], k-nearest neighbor (k-NN) [23], decision tree (DT) [24], naive Bayes (NB) [25], and artificial neural networks (ANNs) [26], for classifying fungal pathogen infections using laser speckle techniques. The authors found that k-NN, DT, and ANN demonstrated strong robustness and high classification performance. It is also worth noting that study [27] presents a system capable of multi-class classification using a CNN. The results were compared to those of an SVM, which proved significantly more resource-intensive and unsuitable for processing the required data volume. Thus, artificial neural networks (ANNs) represent a suitable option for classifying microorganisms based on speckle imaging.

Machine learning (ML) is widely applied to speckle data analysis. In [28], a multilayer perceptron (MLP) [26] with a single hidden layer is used, where the positive part of the DC-normalized intensity spectrum is fed into the network to predict the minimum inhibitory concentration (MIC) of antibiotics for a given bacterium. A similar approach is adopted in [29], but with dynamic holographic laser speckle imaging (DhLSI), which uses an intense reference beam to enable holographic acquisition of speckles. This enhances sensitivity compared to direct imaging, particularly at low bacterial concentrations where scattering signals are weak. DhLSI also holds potential for direct antimicrobial susceptibility testing (AST) without prior bacterial isolation.

In [30], bacterial detection and Gram classification are performed using a two-stream convolutional neural network (CNN) [31]. One stream processes spatial speckle image sequences, while the other uses optical flow-derived grayscale heatmaps to capture temporal dynamics. Outputs from both streams are combined via global average pooling before final classification.

In [32], raw speckle data were analyzed using a 3D CNN combined with an LSTM-based RNN to extract spatiotemporal features [33]. However, the study focused on complex networks without detailing data types, resulting in unstable outcomes. In contrast, our approach performs pre- and postprocessing of speckle data and uses a simpler neural network, highlighting the critical role of processing alongside ML and yielding more stable results.

Accordingly, in the current study, artificial neural networks (ANNs) were utilized, taking advantage of their massively parallel structure and strong generalization capabilities. Additionally, ANNs can reject ambiguous patterns, further improving classification performance [26].

## 2. Materials and Methods

### 2.1. The Experimental Setup

The experimental set-up consists of a 10 Mpix CMOS camera for image capture. Camera “uEye UI-1492LE-C” (IDS Imaging Development Systems, GmbH, Obersulm, Germany); lens “JHF16M-MP2”, manufacturer (Space, Inc., Otawara, Japan) and exposure time—one second. The images were taken at 20-s intervals in bacterial experiments and at 4-s intervals in fungal experiments with corresponding sampling frequencies of 50 mHz and 250 mHz, respectively. The rationale and detailed calculations regarding the sampling frequency of microorganism signals are presented in [34]. An expanded 658 nm laser beam was used to generate laser speckles, ensuring uniform illumination across the entire Petri dish (Figure 1). Laser diode “LP660-SF60” (Thorlabs, Inc., Newton, NJ, USA), and laser spot diameter 12 cm.

### 2.2. Experiment Description

For this study, *E. coli* and the antibiotics to which it reacts were chosen. For training and validation sets, we used the following: amoxicillin-clavulanic acid 20/10 µg (30 µg), cefotaxime 5 µg, ceftazidime 10 µg, ampicillin 10 µg, piperacillin-tazobactam 36 µg, ertapenem 10 µg, sulfamethoxazole 23.75 µg/trimethoprim 1.25 µg, ciprofloxacin 5 µg, meropenem 10 µg, chloramphenicol 30 µg, gentamicin 10 µg, and amikacin 30 µg. For the test set, other experiments used imipenem 10 µg and gentamicin 10 µg. The experiments were performed in a separate room, in an incubator at 37 °C. The *E. coli* suspension was prepared in saline to the density of a 0.5 McFarland turbidity standard. Culturing on Petri dishes was performed according to the EUCAST standard procedure—bacteria were inoculated on Mueller–Hinton agar, and antibiotic discs were placed on the surface [35].

Clinical isolates of Aspergillus niger fungi were also chosen for the study. Aspergillus niger, as a mold, was incubated at temperatures ranging from 25 to 30 °C. Inoculation of the fungi on the medium was performed according to the EUCAST standard procedure [36]. The data were obtained under the supervision of Aigars Reinis at the Paul Stradins Clinical University Hospital Laboratory.

### 2.3. Preprocessing and Data Arrangement

Before the realization of the neural network, the following preprocessing algorithms were performed.

Given the known dimensions of the Petri dish and antibiotic tablets, the speckle image scale can be converted from pixels to micrometers. The bacterial size is also known, and spatial comparisons become possible. The original spatial resolution of the images is 16 µm per pixel. Given that the size of an *E. coli* bacterium is 0.4–0.8 µm by 1–3 µm, the area between pixels is 107–640 times the size of one *E. coli* bacterium.

In various studies on spatial or spatiotemporal speckle contrast [37,38] and correlation-based analysis [39], speckle images are divided into N × N pixel sections, with each segment processed individually. Study [39] evaluated different section sizes and identified an optimal configuration for their objectives. Image segmentation, therefore, is a widely adopted preprocessing step in speckle-based techniques.

For each experiment, the entire field (the 90 mm diameter Petri dish used for bacterial experiments) was divided into N × N pixel sections, with N = 10 pixels chosen for analysis, which corresponds to 160 µm × 160 µm. The area under such windows is 10.7 thousand to 64 thousand times larger than the size of one *E. coli* bacterium. Accordingly, using this method, we observe not individual bacteria but the behavior of signals caused by the vital activity of many bacteria in each considered area of space.

Two types of algorithms were applied to each of the N × N sections.

The first algorithm involves the averaging of speckle intensities over a spatial window of NxN pixels, followed by the use of the signal envelope to avoid the influence of local temporal transient spikes [40]. A moving root-mean-square technique or another similar algorithm can be used for this purpose:(1)$$Env[n]=\sqrt{\frac{1}{N}\sum_{k=n-N+1}^{n}sig[k{]}^{2}}$$
where *N* is the length of the window, *n* is the current sample, and *k* is the index running inside the window. Accordingly, *N*—the length of the window—is responsible for the degree of signal smoothing. To avoid outliers when performing the RMS technique, the extreme values can be truncated, as is carried out by adopting the truncated mean technique [41].

Both procedures help reduce temporal and spatial noise. An increase in signal values occurs when bacteria enter the considered spatial window (regardless of whether they are growing or have already stopped). The signal continues to increase as the bacteria are still growing. When the bacteria stop growing (due to lack of nutrients or due to the antibiotic action), the signal does not decrease because there is a large number of bacteria that scatter light in the considered spatial window. However, the signal might significantly slow down or even stop increasing; that is, this algorithm identifies the presence of bacteria but cannot determine their activity status (growing or not).

The second algorithm, referred to as the “Subpixel Correlation Algorithm”, is described in detail in our previous studies [17,18,42]. This algorithm is sensitive to changes in speckle intensity or displacements of speckle patterns. It involves converting a sequence of images into signals proportional to displacements caused by the activity or growth of microorganisms. Here, we provide only a brief summary; more details can be found in our earlier research:

(1) A two-dimensional normalized correlation was performed between consecutive NxN image fragments throughout the experimental field [16]. (2) The value characterizing the changes between consecutive frames was determined by the offset at the location of the correlation peak. To achieve a more accurate offset, interpolation was performed around the peak [43]. (3) Offsets obtained between each pair of adjacent frames were accumulated and converted into a time signal. To avoid local transient spikes, a signal envelope was applied [40], (Equation (1)). Accordingly, the signal is the spatial displacement vector. The x and y components of displacement were considered, and no significant differences between the x and y components were found, so one component (the x component) was used to increase the speed of analysis [44].

An increase in signal values occurs when there is an increase in activity (due to bacterial or fungal growth), while a decrease in signal occurs either due to nutrient depletion or the action of antibiotics or antifungal drugs. This algorithm (laser speckle imaging with sensitive subpixel correlation analysis), in contrast to the first algorithm (“Spatial averaging”), identifies the activity status (growing or not).

The cause of the pattern shift is obviously that as the activity of microorganisms increases, stronger microvibrations of the surface on which the bacteria grow occur, which is recorded as a shift. The microorganism growth also causes pattern change and, as a consequence, decorrelation [39,45,46].

The algorithms are applied from the beginning to the end of the experiment. Both types of data provide information about the formation of inhibition zones [18,42].

The data arrangement process before using the classifier is described below:

The data for each experiment are represented as a three-dimensional matrix: L × M × T, where L and M are the numbers of N × N spatial windows along the x and y axes (approximately 200 by 300 windows), and T is the number of points along the time axis. In different experiments, the time duration varies between 15–20 h, and the time between frames is 20 s for bacteria and 4 s for fungi.

It is also worth noting that the subpixel correlation technique allows for the detection of submicron bacterial events (such as the formation of an inhibition zone) earlier than the averaging algorithm [42]. However, in situations where bacteria outside the inhibition zone cease to grow, the contrast between the inhibition zone and the bacterial zone (using the subpixel correlation method) will be weak, while the spatial averaging algorithm will show better contrast and, accordingly, may provide additional information (Figure 2, time 20 h). Based on these considerations, both types of data should be examined in more depth for classification purposes. In the bottom row, second image from the left, a “point center” appears in the lower area with circles radiating outward. This effect is observed due to the spiral method used to apply bacteria on the agar surface, creating circular or spiral growth patterns. Since the bottom row images (Subpixel Correlation data) are proportional to activity, and this subimage is around 5 h—near the peak of bacterial activity, this red-yellow pattern highlights the microbiological effect of bacterial activity but not an optical effect.

Figure 2 shows a 35 × 40 mm region of a 90 mm Petri dish, with each analysis segment measuring 160 × 160 µm.

### 2.4. Analysis of Signal Behavior at Varying Radii from the Antibiotic Disc

After bacteria are placed in a nutrient medium, they begin to grow, multiply, and demonstrate “activity”. When the antibiotic is placed on the medium with bacteria, it starts to form an inhibition zone around itself that is nearly circular in shape and grows over time. Bacteria grow across the entire Petri dish, but when the substance released by the antibiotic reaches them, their growth stops in that area (along the radius from the antibiotic disk).

Consider the behavior of the first type of data (from the “Spatial averaging” algorithm) under these conditions. The formation of an inhibition zone is characterized by the cessation of growth. That is, data from the inhibition zone before the antibiotic’s influence show the same behavior as data outside the zone. However, when the antibiotic begins to affect the measured area, the data behavior changes. The data values stop increasing. Therefore, the closer the distance to the antibiotic, (1) the earlier the cessation of growth in the data values will occur and (2) the lower the achieved value will be compared to values farther from the antibiotic. In other words, the shape of the signals at the same distance from the antibiotic (at the same radius) should be similar. Considering that these values are lower than those outside the inhibition zone and are more sensitive to noise, it is advisable to take the median of the data over each radius as it moves away from the center (from the antibiotic) [40,41,47] (Equation (2)). Figure 3 demonstrates the behavior of these data inside and outside the inhibition zone. In the top graph of Figure 3, as the distance from the center increases, the cessation or significant slowdown of data value growth, indicating the formation of an inhibition zone at that location, occurs progressively later. Outside the inhibition zone (Figure 3, bottom graph), the cessation or significant slowdown of growth occurs much later and simultaneously across different radii. In some cases, growth ceases completely, in others, there is still a slight tendency to grow, but weaker than at the initial stage. This indicates that some weak dynamics might still be present.

The *y*-axis shows speckle intensity in dB, highlighting both small and large variations.

Consider the second type of data: the signal envelope obtained after applying the subpixel correlation algorithm, which is proportional to changes in speckle intensity or displacements of speckle patterns (the signal values are proportional to the accumulation of peak shifts over time). In this case, the inhibition zone is characterized by a decrease in signal values (indicated by lower signal values—reduced activity following peak signal levels).

To clearly demonstrate the signal’s behavior, as in the previous method, a median of the signal envelopes is calculated over each radius moving outward from the antibiotic disc. This approach makes even low signals—characteristic of the inhibition zone compared to the high values in the active growth zone—more noticeable [42]. This means that by using a median of signal envelopes for each radius, the resulting curves will have reduced noise (Equation (2)).(2)$$\bar{Env[r,n]}=\left[{median}_{\left\{over\;m_{r1}\right\}\;}(Env[m_{r1},n]),\ldots ,{median}_{\left\{over\;m_{rk}\right\}\;}(Env[m_{rk},n])\right]$$
where $m_{rk}$ is the number of signal envelopes at a given distance/radius (*k*) from the center. Consider the signal envelopes within the inhibition zone across several radii. As the distance from the antibiotic increases, the signal drop—indicating the formation of an inhibition zone at that location—occurs progressively later. This delay reflects the time at which the inhibition zone appears at each respective location.

When considering the signal envelopes for radii outside the inhibition zone—in the active bacterial growth zone—the signals will be very similar to each other. The drop occurs later and almost simultaneously, likely due to nutrient depletion (Figure 4). In contrast, when antibiotics are placed without bacteria, no such behavior is observed [42]. In both cases, in the first hours, a certain decline in activity is observed. This is caused not by the behavior of the microorganisms but by the drying out of the agar media and was discussed in [48].

In summary, both data types show a clear distinction between signals inside and outside the inhibition zone. However, the signal processed with the subpixel correlation method appears cleaner and sharper, making it more effective for distinguishing between zones.

It is also worth noting that in the demonstrated experiment, the difference between signals after the subpixel correlation method is clearly visible in the 4–10 h interval. For the “Spatial Averaging Algorithm”, the differences become noticeable around 4–4.5 h. Up to 4 h, there is almost no difference between the signals. After 9–10 h, the distinction is observed only in the first type of data (from the “Spatial Averaging Algorithm”). However, these type of data are more prone to noise, making it more challenging to make definitive decisions. Similar behavior is observed in other experiments.

Accordingly, it makes sense to evaluate the classification in the interval where the signals are most distinctly separated. This will be carried out using two approaches:
(1)“Long-term” approach: The signal section where the difference is clearly visible (e.g., a 5-h window) is processed as one indivisible segment.(2)“Short-term” approach: The data are divided into smaller time intervals (e.g., 1-h segments). The first approach is expected to yield better results, but this will only be apparent after the entire 5-h interval has passed.

The second approach may provide worse results but offers the advantage of classification at shorter time intervals (e.g., 1 h), which can be processed nearly in real time.

### 2.5. Neural Network for Classification

The main goal of using a neural network in this study is to classify two distinct classes:(1)Zones of active bacterial growth(2)Inhibition zones formed around antibiotic discs

Additionally, the neural network will track changes in the size of the inhibition zones over time. Below, we will describe the key considerations for designing the neural network in accordance with these classification goals.

#### 2.5.1. Neural Network Considerations

For classification applications, the convolutional neural network (CNN) is commonly used and has demonstrated strong performance with images [31]. However, since we are working with sets of 1D temporal signals, we use a regular multilayer perceptron (MLP) [26]. The perceptron convergence algorithm is adaptive and simple to implement. In future work, types of neuron networks will be analyzed to provide improved classification results.

The basic features of MLP:

(1) The model of each neuron in the network includes a nonlinear (and differentiable) activation function [49]; (2) The network contains one or more hidden layers; (3) The network exhibits high connectivity defined by its synaptic weights; (4) The backpropagation method is used for updating weights and calculating the gradient [50].

MLPs are universal approximators and can approximate any function [51]. Study [52] claims that an MLP with two hidden layers is sufficient to generate classification regions of any desired shape. Adding neurons to the second hidden layer allows the network to achieve results with a relatively small number of neurons in the first hidden layer [53]. Study [54] applied a “transformative optimization” method to compare single-hidden-layer and two-hidden-layer networks. In nine out of ten cases, the two-hidden-layer network outperformed the single-hidden-layer network.

A small number of neurons in the hidden layers can lead to underfitting, where the model struggles to adequately detect existing patterns. Conversely, too many neurons in the hidden layers can cause overfitting, as the training data may not be sufficient to properly train all the neurons.

The general rules of thumb [55] suggest the number of hidden neurons should fall between the sizes of the input and output layers, approximately 2/3 of the input size plus the output size, and not exceed twice the input size.

Alternative approaches also exist. Study [56] focuses on cases where a suitable neural network can be selected automatically, providing formulas for determining the upper bounds on the number of hidden layers and neurons in the network design. These approaches can serve as a helpful starting point and guide for selecting both the number of layers and the number of neurons per layer when designing the network.

#### 2.5.2. Database and Neural Network Parameters

Choosing the optimal network topology is challenging due to the variety of architectures and goals, so search algorithms are used to tune parameters.

Following recommendations from the literature, networks with 1, 2, and 3 hidden layers were tested, along with variations in the number of neurons per layer. Figure 5A shows the results, displaying the mean squared error (MSE) as a function of epochs. The data indicate that networks with 2 or 3 hidden layers exhibit lower MSE compared to networks with only one hidden layer. Moreover, a comparison between networks with 2 and 3 hidden layers reveals almost no significant difference.

As the number of neurons increases from a small initial value, the performance improves, but after reaching a certain threshold, further increases in the number of neurons result in minimal improvement (Figure 5B).

The learning rate was also tested, and Figure 5C shows the results: Increasing the learning rate initially reduces the MSE, but beyond a certain point, instability arises, characterized by an increase in MSE (instead of a decrease) or the occurrence of “jumps” in the MSE values.

Figure 5D illustrates the performance improvement when switching from using the first type of signal (“Spatial averaging”) to the second type (“Correlation subpixel algorithm”), as well as when both signals are used simultaneously. Figure 5D illustrates the impact of using each type individually versus both together, emphasizing the advantage of their combined use. All other images use both data types.

Figure 5E examines the time interval from the inoculation time during which the processing window is located. During the time windows 3–4 h after the start of the experiment, bacterial activity remains high for several hours, and against this backdrop, the effect of the antibiotic is clearly visible. However, as bacterial activity decreases (due to nutrient depletion, etc.), the growth of the inhibition zone induced by the antibiotic also diminishes.

The number of epochs was chosen such that the MSE values nearly plateau, indicating minimal further reduction in error. Figure 5F illustrates why less than 150–200 epochs were used: further training yields no significant improvement and may lead to overfitting-related fluctuations and increased error.

Based on the considerations outlined above, the network parameters are as follows: The multilayer perceptron (MLP) [26] network has two hidden layers. The first hidden layer contains 30 neurons, the second hidden layer contains 20 neurons, and the output layer has 2 neurons, corresponding to the two classes (bacteria or inhibition zone). Learning Rate = 0.03; Momentum = 0.5; Epochs = 200.

The neural network was trained using data from 8 experiments with *E. coli* bacteria, and testing was conducted on data from 1 experiment. Each experiment involved almost the entire surface of a Petri dish, in which 2 antibiotics were placed, creating zones of inhibition around them.

Each experiment contained approximately 80–100 thousand signals of the first type (“Spatial Averaging Algorithm”) and the same number of signals from the second type (using the “Correlation subpixels algorithm”), with each signal having a length of T (15–18 h). The signals were divided into 1-h time windows to generate classification results for each short period, thus increasing the total number of signals by a factor of 15–18. Accordingly, in each experiment, there will be about 1–2 million 1-h signals of the first type and the same number of signals of the second type. That is, 2–4 million signals for classification aims.

Since the study focuses on both bacteria and fungi, the classifier was also tested in an experiment with Aspergillus niger fungi. Our previous study outlines the differences between fungal and bacterial signals [34], and this was taken into account to ensure the signals would align. However, future work will involve separating bacteria and fungi to improve classification performance.

## 3. Results

### 3.1. Classification Results

Following the standards that govern the parameters of the experiment [35], the inhibition zone becomes visible approximately 3–4 h after the start of the experiment and can be seen almost immediately with a noticeable radius. In our previous study [42], we explained the underlying reasons for this and demonstrated how the inhibition zone can be observed nearly from its onset. However, we expect that the classifier will not be able to detect the inhibition zone until 3–4 h into the experiment, since by this time, the concentration of bacteria has not yet reached the level at which it can be detected [42]. Nonetheless, we will test this hypothesis.

For reference, consider the “long-term” classification approach. As described in Section 2.4, the signal segment where the difference is clearly visible is processed as a single, indivisible piece. After evaluating different signal sections of lengths 3–6 h and 12 h, a 5-h segment was selected, corresponding to the time interval of 4–9 h from the start of the experiment. In this period, the classifier yielded the best results. Figure 6 shows the results using only the subpixel correlation algorithm, while Figure 7 demonstrates the improvement achieved when both types of data (obtained using spatial averaging and subpixel correlation algorithms) are used.

Classification results based on a 1-h window will be provided in the next subsection.

The results of the fungal activity classification are shown in Figure 8. The left image presents a representative RGB image captured between the 13th and 14th h of the fungal growth experiment, while the right image displays the classification of fungal activity during this period. No antifungal agents were used in these experiments. Despite the fact that the fungal experiment was conducted without antifungal drugs, there are areas of active fungal growth and areas without fungi (which resemble media with no fungal activity, similar to an inhibition zone).

Application of the classifier to fungal experiments demonstrated the potential for classification capabilities, which will be investigated in the future.

### 3.2. Improving Classification Results

This subsection describes two methods for improving classification results by incorporating knowledge about the antibiotic position and the growth process of the inhibition zone, using signal processing.

#### 3.2.1. Improving Classification Results by Using a Class-Changing Function Based on the Distance from the Antibiotic

Considering the following:(1)The inhibition zones occur around the antibiotic, and the location and size of the antibiotic are known. Although the exact size, shape, radius, and location of the antibiotic are known in each experiment, its position can also be reliably detected using algorithmic tools. There are several approaches to circle detections: (a) based on Hough transform [57], (b) based on random sampling [58], (c) based on edge detection technique [59], (d) based on different intelligent optimization algorithms [60], (e) based on circle properties [61], (f) and various others [62,63]. In ref. [64], we demonstrated that speckle imaging with image processing algorithms can be used not only to detect an antibiotic tablet but also to identify the imprinted name of the antibiotic.(2)The inhibition zone typically forms in a shape close to a circle; the classification result can be improved.

For this purpose, the median of the class values obtained at each radius, as it moves from the antibiotic towards the edges, is calculated for each 1-h time window. The class characterizing the inhibition zone is designated as class 1, while the class representing the area with bacteria is designated as class 2. At a radius near the antibiotic, the median class value will be close to 1 (since this represents the inhibition zone). Conversely, at a radius farther from the antibiotic, the median class value will be close to 2 (since this represents an area with active bacteria). The distance at which the transition from class 1 to class 2 occurs will correspond to the radius of the inhibition zone for each time window considered (Equation (3)).(3)$$\langle Class(r)\rangle =\left[median\;(Class[m_{r1}]),\ldots ,median(Class[m_{rk}])\right]$$
where $m_{rk}$ is the number of class points per radius “*k*” from the center.

In Figure 9, the middle and bottom rows show the class change as a function of radius around two antibiotic discs and the improvement in classification results for one selected time window (top row).

Figure 9 illustrates a method for improving classification results within a 1-h time window (6.2–7.2 h). To better demonstrate its effectiveness, three additional 1-h windows are considered: 4.1–5.1 h, 8.2–9.2 h, and 16.4–17.4 h.

Figure 10 illustrates the improvement in classification results by applying a class change function across three different time windows: the beginning, middle, and end.

For the time window 4.1–5.1 (h), total correct detection before improving = 70%: correct bacteria detection = 68%, and correct inhibition zone detection = 74%. Total correct detection after improving = 88%.

For the time window 8.2–9.2 (h), total correct detection before improving = 88%: correct bacteria detection = 86%, and correct inhibition zone detection = 90%. Total correct detection after improving = 92.5%.

For the time window 16.4–17.4 (h), total correct detection before improving = 70%: correct bacteria detection = 69%, and correct inhibition zone detection = 70%. Total correct detection after improving = 83.6%.

Table 1 presents the Performance Metrics Summary before and after correction across all analyzed time windows of the signal. The table includes accuracy (Acc), true positive rate (TPR), false negative rate (FNR), true negative rate (TNR), and false positive rate (FPR). The suffix “Cr” (correction) indicates the corresponding values after correction.

#### 3.2.2. Improving Classification Results Using the Median Signal per Radius

Another approach to improving the classification result, like the previous case, utilizes the same two factors: (1) inhibition zones form around the antibiotic, and the antibiotic’s location and size are known; (2) the inhibition zone typically grows in a shape close to a circle. Based on this knowledge, it is possible to compute the median of the signals over the radius surrounding the antibiotic. As a result, instead of handling a large number of signals, each radius will have only one median signal derived from the subpixel correlation method and another from the “Spatial Averaging Algorithm”. The median signal retains the overall trend of the original signals while suppressing noise [40,41,47], thereby enhancing classifier performance. Additionally, replacing numerous signals with a smaller set of median signals reduces the classifier’s operating time.

The implementation of this approach makes it possible to achieve higher classification performance (Figure 11).

However, if a classification error occurs, the incorrect result will not be confined to a single point in space but will instead appear along an entire specific radius, forming a “ring” shape (Figure 12). If a “ring” is located deep in another class (Figure 12 top right), then it can be easily detected and removed using additional algorithmic tools.

If a “ring” is located at the transition between two classes, detecting the error becomes more difficult. However, this can be performed using neighborhood analysis in the pre-median data.

Table 2 presents the performance metrics summary of this method across all analyzed time windows of the signal. The table includes accuracy (Acc), true positive rate (TPR), false negative rate (FNR), true negative rate (TNR), and false positive rate (FPR).

## 4. Summary and Discussion

In the current study, the formation of inhibition zones is monitored from the time of application of bacteria and antibiotics until the end of the experiment. Thus, the dynamics of zone formation are observed and detected in short time intervals of 1 h to provide results quickly, in a time comparable to the small changes in the growth of microorganisms, rather than after the typical 16–24 h [35]. In contrast, some recent studies classify inhibition zones and active bacterial growth zones using laser speckle techniques combined with correlation methods and classification tools only after the inhibition zones have fully developed, often 24 h post-incubation. For instance, study [39] utilized principal component analysis (PCA) and k-means clustering for the classification of bacterial regions based on correlation values. Although their study demonstrates promising results, our approach differs as it does not rely on individual correlation values at specific times post-incubation. Instead, we analyze complete signal patterns, which vary significantly over time. To address this, we decided to employ neural networks. Neural networks possess the ability to reject ambiguous patterns and exhibit strong generalization capabilities, thereby further enhancing classification accuracy [26].

The subpixel correlation algorithm, which is highly sensitive to changes in activity, serves as an effective monitoring tool. In the inhibition zone, the signal values from the subpixel correlation algorithm decreased dynamically (i.e., as one moves farther from the antibiotic, the decrease occurs later), representing the radius of the inhibition zone. Outside the inhibition zone, the signal decreased much later and simultaneously across different radii. Based on this principle, it is possible to create classifiers between zones. The signals in both zones were clearly distinct from each other. The incorporation of an additional spatial averaging (or median) algorithm, known for its sensitivity to another parameter, speckle intensity, significantly contributed to enhancing the results. Thus, the correct classification reached 88% in the 8.2–9.2 h time window. However, at the beginning of the inhibition zone’s formation, or conversely, many hours after its onset, the percentage of correct classification decreased to 69–70%. Using the proposed methods, (1) by finding the radius at which the inhibition zone ends, the correct classification reaches 92.5% in the 8.2–9.2 h time window, and 78–88% at the beginning of the inhibition zone’s formation or many hours after its onset. (2) Using the median signal per radius, the correct classification reaches 94–99% in all time-windows.

Thus, by understanding the specificity of signals and utilizing an appropriate algorithmic base before and after applying neural networks, high classification accuracy can be achieved.

The proposed technique is applicable for classifying bacterial activity, and the potential could be applicable for fungal activity, which will be studied in more detail in our subsequent work. Future work will focus on distinguishing between bacteria and fungi to achieve even higher performance.

The advantages of laser speckle imaging, enhanced with subpixel correlation methods, make it a powerful tool for in-depth studies of microorganism growth dynamics and microbial resistance mechanisms. This study aims to demonstrate the utility of laser speckle imaging in distinguishing between active and inhibition zones. In the current study, the emphasis is on the classification principles and the algorithmic tools. However, in future work, types of neuron networks will be analyzed to provide improved classification results. By enabling in situ observations, the method facilitates real-time monitoring of microbial responses to various chemicals and potential inhibitors. It could become a valuable tool for studying bacterial and fungal growth dynamics as well as evaluating the efficacy of antimicrobial substances.

## 5. Conclusions

The integration of subpixel correlation analysis with laser speckle imaging provides a more detailed understanding of microorganisms’ activity and inhibition zones, something that cannot be achieved with laser speckle or white-light imaging alone. The subpixel correlation algorithm demonstrated much better results than the spatial averaging algorithm. However, using both of these algorithms together for classification purposes helps achieve even better results. Machine learning algorithms facilitate the automatic classification of inhibition zones and active bacterial areas. To achieve even better classification results, especially in highly noisy environments or cases where inhibition zones are not clearly distinguishable, the types and capabilities of neural networks will be analyzed in greater detail within the context of the tasks demonstrated in this study.

The demonstrated method for distinguishing between inhibition and growth zones of microorganisms enables deeper investigations into the complex interactions within microbial communities.

Based on the obtained results, a statistical analysis will be conducted in future studies to examine the relationship between the size and growth rate of the inhibition zone and the type of antibiotic or antifungal drug used, for the same bacterial or fungal strains. The goal is to develop an algorithm that can predict the radius of the inhibition zone around antimicrobial agents much more quickly than the standard disk diffusion method. Neural networks provide a well-considered and effective solution for such tasks.

The ongoing research aims to expand this classification to zones with varying microbial characteristics, thereby enhancing the method’s utility in both fundamental and applied microbiology.

## Figures and Tables

**Figure 1 sensors-25-03462-f001:**
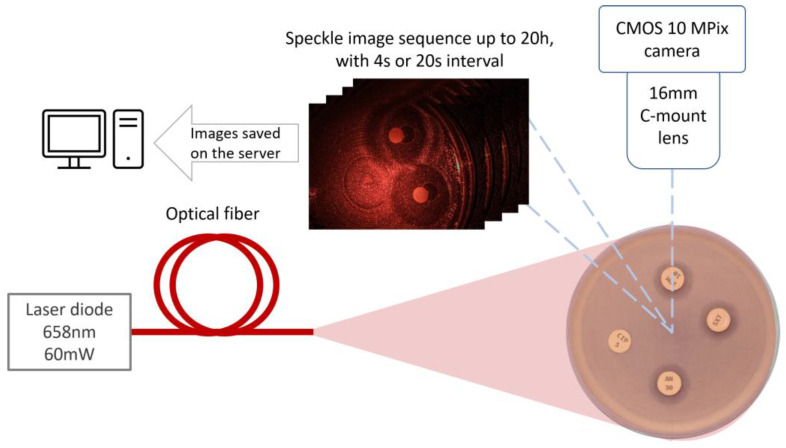
Setup scheme for burst image capturing of bacteria growing process.

**Figure 2 sensors-25-03462-f002:**
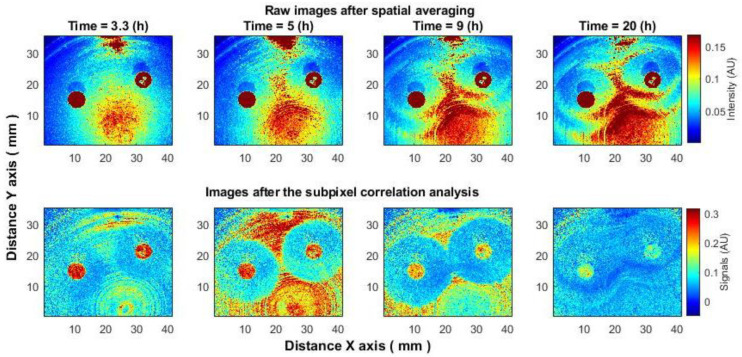
Changes in the inhibition zone as a function of time. Bacteria: *E. coli*, antibiotic CAZ 10 µg (left disc) and TZP 36 µg (right disc). Processed using an averaging (smoothing) filter and contrast enhancement (**top row**) and resulting images after subpixel correlation analysis (**bottom row**).

**Figure 3 sensors-25-03462-f003:**
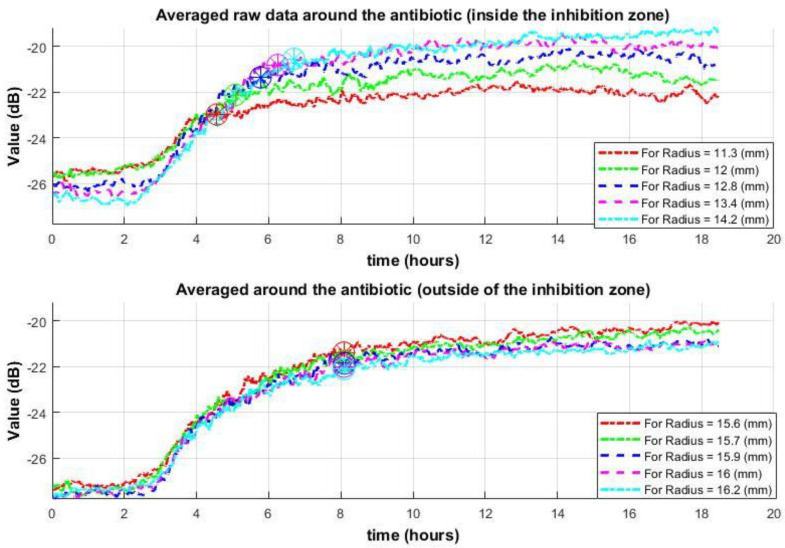
Spatial dynamics of the “Spatial Averaging Algorithm” over time around the antibiotic. The (**top**) graph: data within the inhibition zone, the (**bottom**) graph: data outside the inhibition zone (with active bacteria). Circles mark the points of growth cessation or significant slowdown.

**Figure 4 sensors-25-03462-f004:**
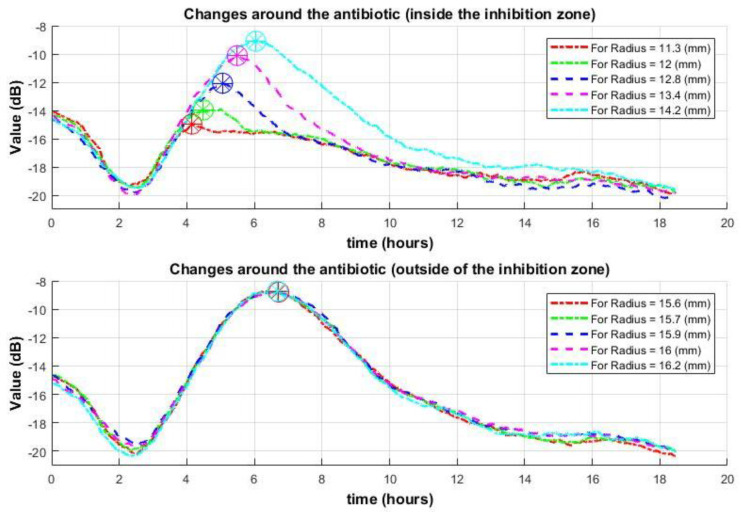
Behavior of the median signal envelopes by radius of the “Subpixel Correlation Algorithm” over time for several different radii around the antibiotic. The (**top**) graph: data within the inhibition zone, the (**bottom**) graph: data outside the inhibition zone (with active bacteria). Circles mark the points of growth cessation.

**Figure 5 sensors-25-03462-f005:**
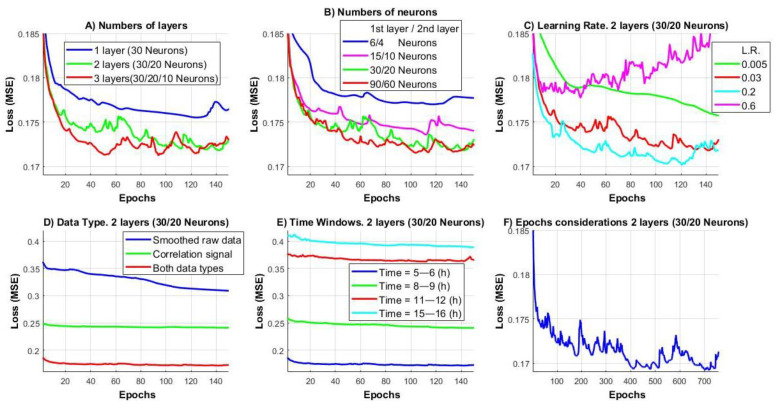
Parameter analysis: number of layers (**A**), number of neurons (**B**), learning rate (**C**), data type (**D**), time window position (**E**), number of epochs (**F**).

**Figure 6 sensors-25-03462-f006:**
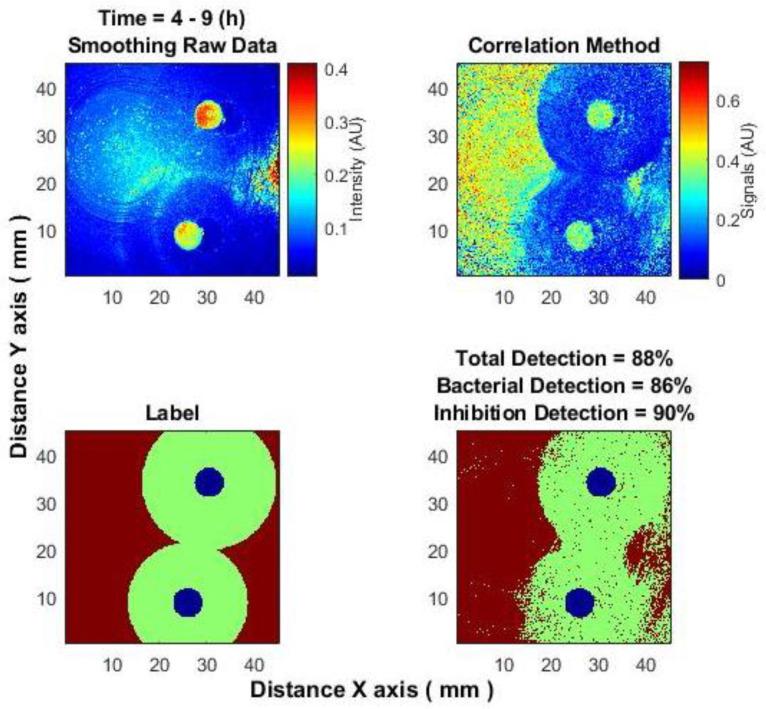
“Long-term” classification: (**top left**)—smoothed raw data (after spatial averaging); (**top right**)—subpixel correlation method; (**bottom left**)—labels; (**bottom right**)—classification result.

**Figure 7 sensors-25-03462-f007:**
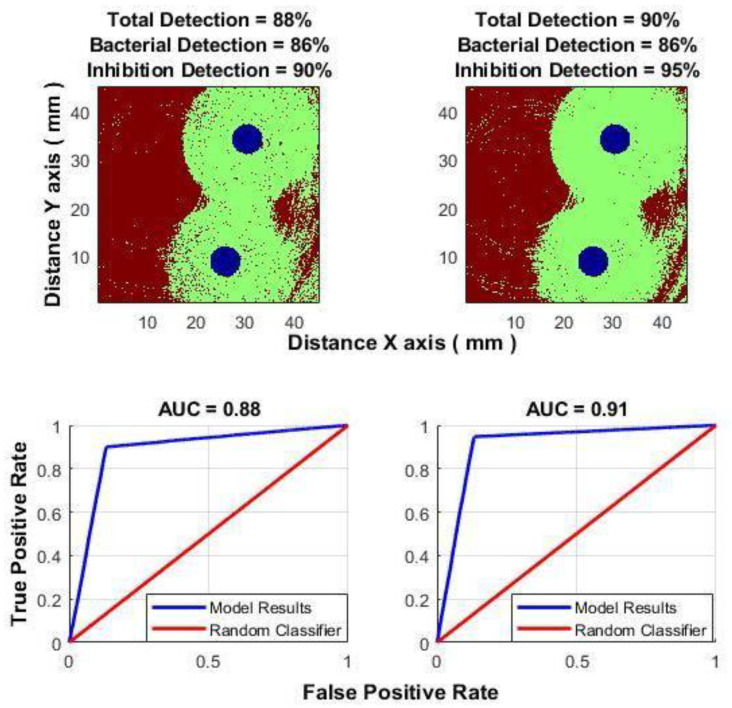
“Long-term” classification: (**top left**)—classification result for subpixel correlation method; (**top right**)—classification result for both methods together: subpixel correlation method and spatial averaging; (**bottom**)—ROC curves accordingly.

**Figure 8 sensors-25-03462-f008:**
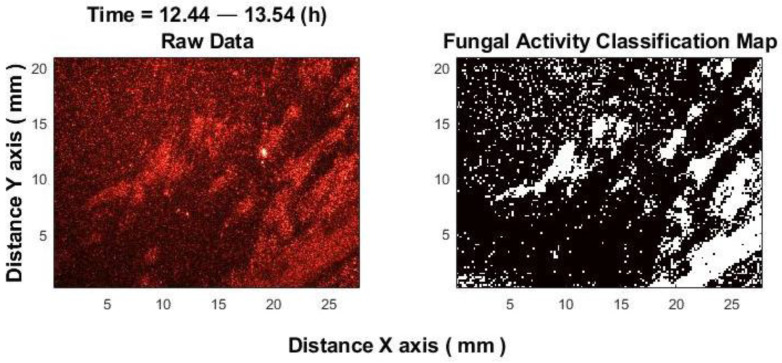
Fungal activity classification. On the (**left**) is the raw signal. On the (**right**) is the result. White color—fungal activity. Black color—no activity.

**Figure 9 sensors-25-03462-f009:**
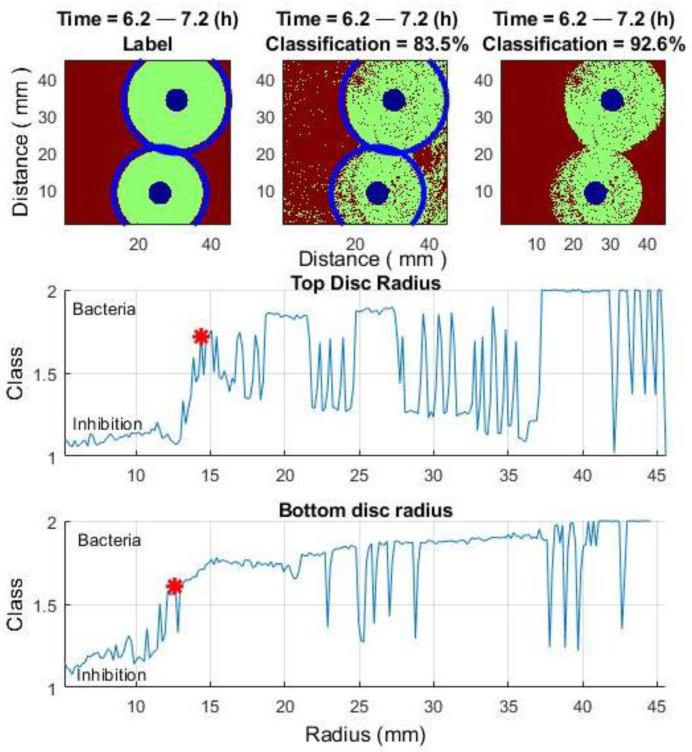
Improving classification results. Time window: 6.2–7.2 h. (**Top row**): (**Left**) image—labeling; (**Central**) image—classification result; (**Right**) image—classification result after applying the class estimation algorithm. (**Middle row**): Class change as a function of radius for the top antibiotic disc. (**Bottom row**): Class change as a function of radius for the bottom antibiotic disc. The red asterisk marks the time of class transition.

**Figure 10 sensors-25-03462-f010:**
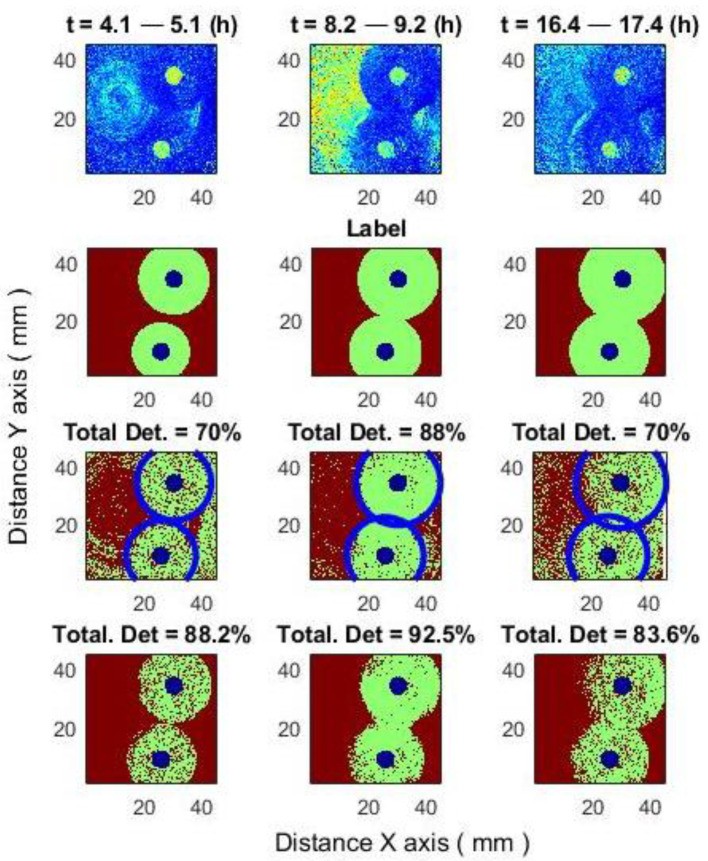
Improving classification results for three different time windows: the beginning, middle, and end.

**Figure 11 sensors-25-03462-f011:**
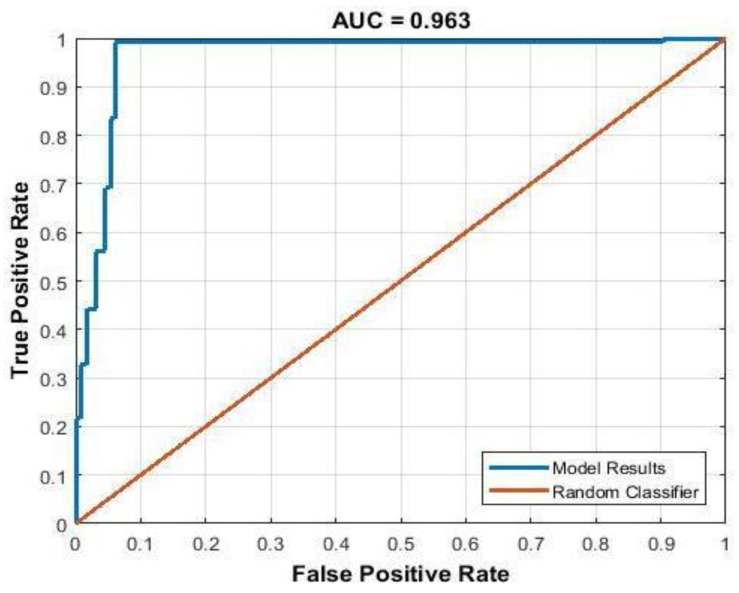
ROC curve of classification using a median signal.

**Figure 12 sensors-25-03462-f012:**
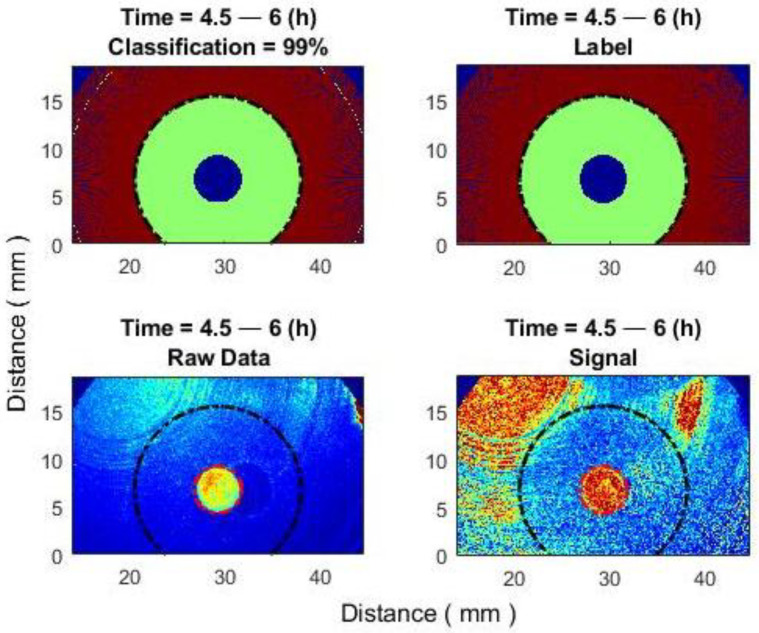
Improving classification results using one median signal per radius. Time window: 4.5–6 (h). (**Top left**)—classification result (example of a classification error: green “ring” (inhibition) is located deep in another brown (active) class); (**top right**)—labels; (**bottom left**)—averaging over an N × N pixel window; (**bottom right**)—subpixel correlation algorithm.

**Table 1 sensors-25-03462-t001:** Performance metrics summary before and after correction.

Time (h)	Acc	Acc Cr	TPR	TPR Cr	FNR	FNR Cr	TNR	TNR Cr	FPR	FRP Cr
4–5	70	88	74	72	26	28	68	98	32	2
5–6	82	92	82	82	18	18	82	99	18	1
6–7	84	93	92	91	8	9	78	97	22	3
7–8	87	93	89	89	11	11	85	98	15	2
8–9	88	93	90	88	10	12	86	98	14	2
9–10	87	93	90	88	10	12	83	98	17	2
10–11	83	95	91	90	9	10	74	100	26	0
11–12	74	78	61	61	39	39	89	99	11	1
12–13	72	79	63	63	37	37	83	99	17	1
13–14	69	80	65	65	35	35	75	99	25	1
14–15	69	81	67	67	33	33	71	99	29	1
15–16	69	82	68	68	32	32	70	99	30	1
16–17	70	83	70	70	30	30	69	99	31	1

**Table 2 sensors-25-03462-t002:** Performance metrics summary for final model evaluation.

Time (h)	Acc	TPR	FNR	TNR	FPR
4.5–6	99	99	1	99	1
6–7.5	98	95	5	100	0
7.5–9	96	93	7	100	0
9–10.5	94	89	11	100	0
10.5–12	94	90	10	100	0
12–13.5	96	93	7	100	0
13.5–15	97	95	5	100	0
15–16.5	96	99	1	91	9

## Data Availability

The raw data supporting the conclusions of this article will be made available by the author upon request.

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
