# Peer review of "Classification of Microbial Activity and Inhibition Zones Using Neural Network Analysis of Laser Speckle Images"

_sensors, 2025, doi:10.3390/s25113462_

Round 1
Reviewer 1 Report
Comments and Suggestions for Authors
see attached

Author Response
Please see the attached author responses.

Reviewer 2 Report
Comments and Suggestions for Authors
Please, see attached report file.

Author Response

(The authors gave the same response as above.)

Round 2
Reviewer 1 Report
Comments and Suggestions for Authors
The authors have addressed all my previous comments and the manuscript is much improved with additional details where required and revisions to text to reduce length.
Author Response
Thank you
Reviewer 2 Report
Comments and Suggestions for Authors
The authors have revised the manuscript titled “Classification of Microbial Activity and Inhibition Zones Using Neural Network Analysis of Laser Speckle Images”. Some of my questions were addressed properly. However, some points still need attention before the manuscript can be considered for final publication.
- Section “1. Introduction”: In their reply-to-reviewers letter, authors have written that they have expanded the introduction section, however many of the suggested papers are still missing. The introduction section is still quite poor and does not correctly frame the work in the context of speckle pattern imaging applied to analyze scattering elements in liquids, which actually one of the latest innovative application of the technique. Author have to include a general overview on the use of speckle pattern imaging to analyze dynamic phenomena. I strongly suggest that they include the following papers that were previously indicated:
-
- DOI: 10.1038/s41467-023-36816-2
- DOI: 10.1109/TIM.2023.3289543
- DOI: 10.1016/j.measurement.2023.113590
- DOI: 10.1088/1555-6611/acde6f
- DOI: 10.1109/JPHOT.2020.3044912
- DOI: 10.1016/j.jqsrt.2020.107496
-
- Section “2.1. The experimental setup”: Vendor name and vendor country of each optical component must be added.
- Section “3. Results”: model accuracy, sensitivity, true positive rate and false negative rate are still missing: please do report them in the text.
- General comment: I still have some doubts regarding the actual need of image segmentation and radius calculation with the use of AI. Can the authors explain (and cite some related references) how this calculus was made before the diffusion of ML/DL models? Can they provide a comparison between performances obtained using simple image processing techniques and those obtained by applying AI?
Author Response
- Section “1. Introduction”: In their reply-to-reviewers letter, authors have written that they have expanded the introduction section, however many of the suggested papers are still missing. The introduction section is still quite poor and does not correctly frame the work in the context of speckle pattern imaging applied to analyze scattering elements in liquids, which actually one of the latest innovative application of the technique. Author have to include a general overview on the use of speckle pattern imaging to analyze dynamic phenomena. I strongly suggest that they include the following papers that were previously indicated:
- DOI: 10.1038/s41467-023-36816-2
- DOI: 10.1109/TIM.2023.3289543
- DOI: 10.1016/j.measurement.2023.113590
- DOI: 10.1088/1555-6611/acde6f
- DOI: 10.1109/JPHOT.2020.3044912
- DOI: 10.1016/j.jqsrt.2020.107496
Dear Reviewer,
In the first review round, you kindly suggested expanding the Introduction section with additional references. We thoroughly revised it, incorporating around 15–20 new citations: both your recommendations and other relevant works, and provided extended background and literature analysis.
In this round, we have added further data and included the following text:
"Study [20] compares various machine learning classifiers, including support vector machine (SVM) [21], logistic regression (LR) [22], k-nearest neighbor (k-NN) [23], decision tree (DT) [24], Naive Bayes (NB) [25], and artificial neural networks (ANNs) [26] for classifying fungal pathogen infections using laser speckle techniques. The authors found that k-NN, DT, and ANN demonstrated strong robustness and high classification performance. It is also worth noting that study [27] presents a system capable of multi-class classification using a CNN. The results were compared to those of an SVM, which proved significantly more resource-intensive and unsuitable for processing the required data volume. Thus, artificial neural networks (ANNs) represent a suitable option for classifying microorganisms based on speckle imaging."
And: “For analyzing speckle patterns containing both dynamic and static regions, study [14] demonstrated that laser speckle correlation can be effectively used to estimate the relative concentrations of static and dynamic scatterers within a sample.”
Regarding your suggestion to further address "speckle pattern imaging applied to analyze scattering elements in liquids," we would like to respectfully note that the current study focuses on solid (agar-based) media, not liquid environments. Therefore, phenomena such as drying of wet powder, turbid fluids like diluted rice milk samples or nanoparticle suspensions, while valuable, lie outside the scope of this work, as they do not address microbial behavior analysis.
Given this, we believe that the references and discussions added during both review rounds adequately cover the relevant domain of our research.
- Section “2.1. The experimental setup”: Vendor name and vendor country of each optical component must be added.
The data have been added to the text of the manuscript:
"The experimental set-up consists of a 10 Mpix CMOS camera for image capture. Camera “uEye UI-1492LE-C”, manufacturer “IDS” (Germany); lens “JHF16M-MP2”, manufacturer “Spacecom” (Japan). Exposure time - one second. Images were taken at 20-second intervals in bacterial experiments and at 4-second intervals in fungal experiments. Corresponding sampling frequencies of 50 mHz and 250 mHz, respectively. The rationale and detailed calculations regarding the sampling frequency of microorganism signals are presented in [34]. An expanded 658 nm laser beam was used to generate laser speckles, ensuring uniform illumination across the entire Petri dish (Fig. 1). Laser diode “LP660-SF60”, manufacturer “ThorLabs” (USA). Laser spot diameter 12 cm."
- Section “3. Results”: model accuracy, sensitivity, true positive rate and false negative rate are still missing: please do report them in the text.
The data have been added to the text of the manuscript (Section 3.2.1):
“Table 1 presents the confusion table before and after correction across all analyzed time windows of the signal. The table includes Accuracy (Acc), True Positive Rate (TPR), False Negative Rate (FNR), True Negative Rate (TNR), and False Positive Rate (FPR). The suffix "Cr" (Correction) indicates the corresponding values after correction.
Table 1. Confusion table before and after correction.”
Please see the table in the manuscript.
The data have been added to the text of the manuscript (Section 3.2.2):
“Table 2 presents the confusion table of this method across all analyzed time windows of the signal. The table includes Accuracy (Acc), True Positive Rate (TPR), False Negative Rate (FNR), True Negative Rate (TNR), and False Positive Rate (FPR).
Table 2. Confusion table of the proposed method.”
Please see the table in the manuscript.
- General comment: I still have some doubts regarding the actual need of image segmentation and radius calculation with the use of AI. Can the authors explain (and cite some related references) how this calculus was made before the diffusion of ML/DL models? Can they provide a comparison between performances obtained using simple image processing techniques and those obtained by applying AI?
Regarding image segmentation, the following text has been added into section 2.3: "In various studies on spatial or spatiotemporal speckle contrast [37,38] and correlation-based analysis [39], speckle images are divided into NxN pixel sections, with each segment processed individually. Study [39] evaluated different section sizes and identified an optimal configuration for their objectives. Image segmentation, therefore, is a widely adopted preprocessing step in speckle-based techniques”
Regarding the radial averaging calculations, signal processing offers well-known noise reduction techniques, including averaging of similar signals [40,41,47]. Since the inhibition zone expands radially, signals at equal distances from the antibiotic center are expected to be similar and can thus be averaged ( or performed median ) to suppress noise. This approach was implemented both as a preprocessing step prior to neural network input (3.2.2) and as a postprocessing step based on the network's output (3.2.1). In both cases, the neural network performs only classification, while the radial computations are performed separately.
Numerous methods have been proposed for detecting inhibition zones in white-light images, ranging from classical image processing and computer vision techniques [3s, 4s] to AI-based approaches [1s, 2s]. However, for speckle images characterized by granular, irregular patterns AI methods are generally preferred [20].
To compare traditional image processing with AI-based approaches, one can contrast our previous study [42] with the current work. In the earlier study, segmentation and subpixel correlation were used to visualize inhibition zones and regions of bacterial activity. The proposed activity detection-oriented method gave good results compared to raw speckle data, visualizing zones at an earlier stage and coping with such limitations as shadows from objects, low contrast and light reflections, indistinct zone boundaries.
Simple thresholding methods were also tested but proved unreliable due to high signal variability inherent to bacterial speckle data. Since white-light image-based methods assume more uniform image characteristics, they are less applicable here. Given the complexity and variability of speckle patterns, we adopted an AI-based classification approach, in line with many studies in this domain.
The following explanatory text has been added to the “1. Introduction”:
"Study [20] compares various machine learning classifiers, including support vector machine (SVM) [21], logistic regression (LR) [22], k-nearest neighbor (k-NN) [23], decision tree (DT) [24], Naive Bayes (NB) [25], and artificial neural networks (ANNs) [26] for classifying fungal pathogen infections using laser speckle techniques. The authors found that k-NN, DT, and ANN demonstrated strong robustness and high classification performance. It is also worth noting that study [27] presents a system capable of multi-class classification using a CNN. The results were compared to those of an SVM, which proved significantly more resource-intensive and unsuitable for processing the required data volume. Thus, artificial neural networks (ANNs) represent a suitable option for classifying microorganisms based on speckle imaging. "
[1s] F. Fough, G. Janjua, Y. Zhao and A. W. Don, "Predicting and Identifying Antimicrobial Resistance in the Marine Environment Using AI & Machine Learning Algorithms," 2023 IEEE International Workshop on Metrology for the Sea; Learning to Measure Sea Health Parameters (MetroSea), La Valletta, Malta, (2023).
[2s] Pascucci, M., Royer, G., Adamek, J. et al., "AI-based mobile application to fight antibiotic resistance," Nat Commun 12, 1173, (2021).
[3s] Alonso CA, Domínguez C, Heras J, Mata E, Pascual V, Torres C, Zarazaga M. Antibiogramj: A tool for analysing images from disk diffusion tests. Comput Methods Programs Biomed. 2017
[4s] H. S. Bhargav, S. D. Shastri, S. P. Poornav, K. M. Darshan and M. M. Nayak, "Measurement of the Zone of Inhibition of an Antibiotic," 2016 IEEE 6th International Conference on Advanced Computing (IACC), Bhimavaram, India, 2016.
